# Validation of a Multiplex Molecular Macroarray for the Determination of Allergen-Specific IgE Sensitizations in Dogs

**DOI:** 10.3390/vetsci11100482

**Published:** 2024-10-07

**Authors:** Thierry Olivry, Ana Mas Fontao, Martina Aumayr, Natalia Paulenka Ivanovova, Georg Mitterer, Christian Harwanegg

**Affiliations:** 1Nextmune AB, Riddargatan 19, SE-114-56 Stockholm, Sweden; 2Nextmune Spain, Valentin Beato 24, 28037 Madrid, Spain; ana.mas@nextmune.com; 3MacroArray Diagnostics, Lemböckgasse 59, 1230 Vienna, Austria; aumayr@macroarraydx.com (M.A.); ivanovova@macroarraydx.com (N.P.I.); mitterer@macroarraydx.com (G.M.); harwanegg@macroarraydx.com (C.H.)

**Keywords:** allergen, allergy, atopic dermatitis, dog, IgE, molecular allergology, serodiagnosis, serology

## Abstract

**Simple Summary:**

To determine which allergens allergic dogs are sensitive to, veterinarians can use either skin or blood tests with allergen extracts. However, these extracts can be hard to standardize and may not contain important proteins at high enough levels. Using molecular allergens can offer certain benefits. In this paper, we introduce the Pet Allergy Xplorer (PAX), the first serum test designed and validated for dogs using principles of molecular allergology. This test, based on a leading platform for allergic humans, is accurate and reproducible and provides unique advantages compared with traditional extract-based tests.

**Abstract:**

Detecting IgE sensitizations in the serum of allergic dogs is commonly performed using allergen extracts, but these are difficult to standardize. This article details the development and validation of the Pet Allergy Xplorer (PAX; Nextmune, Stockholm, Sweden), the first multiplex macroarray for the detection of IgE sensitization in dogs using allergen extracts and molecular components; the PAX is derived from the Allergy Xplorer (ALEX^2^; MacroArray Diagnostics, Vienna, Austria). The selection of allergens, cartridge processing, strategy for identifying and blocking IgE directed against cross-reactive carbohydrate determinants (CCDs), and the method used for determining the positivity threshold are described. The validation of the PAX included evaluations of the specificity of its anti-IgE monoclonal antibody, specificity of IgE binding to target allergens, assay precision, and internal consistency. Additionally, the influence of possible confounding factors, such as sample type, the influence of hemolysis, lipemia, bilirubinemia, and elevated CCD-IgE, was tested. Finally, the sensitization rates of 23,858 European dogs to 145 environmental and *Hymenoptera* venom allergens were summarized. The PAX is accurate and reproducible and has a unique CCD-detection and blocking strategy; its molecular allergens offer a unique window on allergen cross-reactivity.

## 1. Introduction

In human and animal patients, allergen immunotherapy (AIT) aims at preventing the relapse of clinical signs after further allergen contact [1]; it is also the only intervention known to have the potential to prevent new allergen sensitizations [2]. To select allergens for inclusion in AIT formulations, clinicians rely on a combination of history, clinical signs, and determination of IgE sensitizations [3]. In veterinary medicine, the latter normally involves the use of allergen-specific intracutaneous (intradermal or percutaneous [i.e., “prick”]) or IgE serological testing using crude whole allergen extracts. Unfortunately, allergen extracts for testing are difficult to standardize. This extract heterogeneity is best exemplified with extracts of *Dermatophagoides* house dust mites used for human and animal patients, which have been shown to be heterogeneous in their composition and content of important allergens [4,5,6,7]; consequently, testing with such extracts can lead to different results depending upon the extract used.

Molecular allergology, also known as “component-resolved diagnostics, CRD” or “precision allergy diagnostic applications, PAMD@,” involves the testing, usually serological, of single molecular components in addition to, or in replacement of, crude extracts [8]. In humans, molecular allergology improves testing sensitivity and specificity, as each allergen “spot” contains 100% of a single allergen rather than a crude mix of proteins, with few of them being relevant allergens [8]. The *Dermatophagoides farinae* (Der f) house dust mite can again serve as an example: indeed, a Der f extract is expected to contain all all the 10,684 proteins encoded in its genome of which there are, so far, only 37 proteins (or 0.34%) officially recognized as allergens for humans (www.allergen.org; page last accessed 28 June 2024) [9]. A similar issue of low major allergen content also exists in veterinary medicine, as a Der f extract for animal use was reported only to contain between 1 and 2% each of Der f 1, Der f 2, Der f 15, and Der f 18, the main Der f allergens targeted by IgE in dogs with atopic dermatitis (AD) [10]. Other known advantages of molecular allergology are the easier identification of allergen cross-reactivity and primary sensitizing allergens, the possibility of predicting clinical evolution or clinical signs with the detection of IgE sensitization to some specific components, and a more logical selection of allergens for AIT [11].

In humans, IgE serological testing with molecular allergens can be performed either in singleplex or multiplex mode: in the former, allergists request a test for single individual allergens; in the latter, many relevant allergens are spotted on a solid matrix and tested in a single run [11]. The Allergy Xplorer (ALEX^2^, MacroArrray Diagnostics, Vienna, Austria) is a multiplex macroarray used to test sera from human allergy-suspected patients for the determination of IgE sensitizations to about 300 allergens. In its current form, it has about two-thirds of individual molecular allergens (or components) and one-third of crude extracts, which are selected for allergen sources that have incompletely characterized molecular components. The performance of the ALEX^2^ is comparable to that of other multiplex or singleplex molecular tests [12,13,14,15,16,17,18,19,20]

In this paper, we describe the development and validation of the Pet Allergy Xplorer (PAX, Nextmune, Stockholm, Sweden), a quantitative allergen-specific IgE serological test for dogs. This test employs a similar multiplex allergen macroarray and methodology as the ALEX^2^ for humans. For the sake of brevity, the review of PAX results on European dogs tested in 2023 will be limited to environmental allergens with a deeper focus on those of *Dermatophagoides* house dust mites (HDMs).

## 2. Materials and Methods

### 2.1. Canine Sera

During the PAX validation stage in 2022, we used 1019 archived sera from allergy-suspected dogs that had been sent for allergen-specific IgE serological ELISA testing (Next+, Nextmune, Madrid, Spain). We also tested 123 archived sera from laboratory beagles that had been collected during previous research projects between 2018 and 2023; the collection of blood from normal and mite-sensitized laboratory beagles had been approved beforehand by different institutional animal care and use committees. Finally, we used results from 23,858 additional sera from allergy-suspected dogs that had been submitted to one of the three European PAX testing laboratories in 2023. As such testing was part of the routine allergy workup by practicing veterinarians, the review by an ethics panel was not needed for these pet dogs.

### 2.2. PAX Design and Processing

#### 2.2.1. Allergen Selection

We aimed to design the PAX similarly to the ALEX^2^, with around one-third of allergen extracts and two-thirds of molecular components. The selection of allergens for the PAX for dogs began with those offered on the ALEX^2^ after postulating that if a protein is allergenic in humans, it would likely also be so in animals. Indeed, not all proteins from an organism are allergenic—as discussed above for Der f allergens—since they must possess unique features (e.g., three-dimensional structure, enzymatic properties [see section A07 in [11]], or a genetic sequence prone to transcription infidelity) [21] to render them immunogenic and targetable by IgE. From the ALEX^2^, we first removed extracts and components deemed irrelevant for dogs (e.g., food allergens from exotic fruits, shrimp, etc.). We then added allergens shown to be sensitizing dogs, but that were missing from the ALEX^2^. Examples of the latter are the Der f extract and its components Der f 15 and Der f 18, as well as several food allergens, including Bos d 7 (bovine IgG) [22], Gal d 9 (chicken beta-enolase), or Sal s 2, 3, 4, 7, and 8 (Atlantic salmon’s beta-enolase, aldolase, tropomyosin, creatine kinase, and triosephosphate isomerase 1, respectively) [23,24].

In some cases, we substituted the allergen from the species of interest with one from another species. This substitution was made, for example, when the amino acid sequence of the allergen from the original species was incomplete. Nevertheless, before making such a substitution, we ensured either that the proteins from both species shared a high degree of sequence identity or that the A-RISC algorithm predicted a high level of cross-reactivity between these heterologous allergens [25,26]. Most components were recombinant proteins produced in *Escherichia coli* bacteria or *Pichia pastoris* yeast, while few others were purified native allergens.

All extracts and molecular components were coupled to latex nanobeads before being machine-spotted on a nitrocellulose membrane, which was cut and fitted to a small 5.5 by 1.5 cm cartridge (Figure 1) using the same automated process as for the ALEX^2^ (MacroArray Diagnostics, Vienna, Austria).

All allergens on the cartridge were standardized based on their biological activity, which was verified at several checkpoints. At the time of allergen coupling to nanobeads, the coupling method and allergen concentration were validated for the batch’s production. After membrane spotting, additional tests were performed using various positive human and canine sera to confirm the spotting precision.

The list of allergens included in the PAX version 22.2 used in 2022 and 2023 can be found in Appendix A. Altogether, in this version of the PAX for dogs, there were 247 allergen spots (75 individual extracts, two mixes of two extracts, 169 individual components, and one two-component mix), of which environmental aeroallergens occupied 132 spots (53.4%; 39 individual extracts, two two-extract mixes, and 91 components). There were also 13 spots for *Hymenoptera* venoms (5%; five individual extracts and eight individual components), 98 food allergens (39.7%; 31 individual extracts, 66 individual components, and one two-component mix), and two cross-reactive carbohydrate determinant (CCD) detectors and their respective non-CCD controls (all individual components).

#### 2.2.2. Cartridge Processing

As for the ALEX^2^, the sera were processed on a MAX45K (MacroArray Diagnostics), an automaton that can complete the testing of 50 sera in 4 h. Briefly, all canine sera were diluted 1:5 in the PAX sample diluent containing a proprietary mix of proteins carrying CCD. This serum-CCD blocker mix was then incubated on the cartridges for 1.5 h before being rinsed three times with the PAX Washing Solution. For the detection of canine IgE, the PAX uses alkaline phosphatase (AP)-labeled 5.91 monoclonal antibodies previously shown to uniquely recognize an epitope in the Cε2 domain of dog IgE, but not IgG, IgM, or IgA [27]; this PAX Antibody Solution was incubated for 20 min. After five rinses, the PAX Substrate Solution was added and stopped after 9 min. A complementary metal oxide semiconductor (CMOS) sensor reads the colorimetric reaction. Each batch of 1000 cartridges was calibrated to enable results to be matched to a 5-dilution standard curve of dog IgE and expressed in ng/mL of allergen-specific IgE.

#### 2.2.3. Cross-Reactive Carbohydrate Determinant Blocking Strategy

The strategy for detecting and blocking CCD-IgE in the PAX was expanded compared with that of the ALEX^2^. As mentioned above, all sera were diluted with a buffer containing a proprietary mix of proteins containing CCD motifs. Each cartridge contained four spots occupied by two different human proteins that either have or do not have one or both known plant CCD epitopes for humans and dogs [28]. These two “CCD detectors” (recombinant human lactoferrin produced in rice [L4040, Sigma-Aldrich, St Louis, MO, USA], and the MUXF3 complex glycan coupled to human serum albumin [Proglycan, Vienna, Austria]) allow for the evaluation of the efficiency of the initial CCD-IgE block. If, after this block, the specific IgE (sIgE) value for one or both “CCD detectors” was 100.00 ng/mL or higher (a value that our preliminary studies had shown to be associated with numerous sensitizations to plant extracts), the serum was blocked a second time with a 40.00 µg/mL concentration of another proprietary CCD-expressing protein before being retested. If the IgE level against one or both CCD-bearing detectors again was above the positivity threshold after that second CCD-IgE block, the veterinarian is informed of the possible lack of clinical relevance of any sIgE values against plant pollens, plant food extracts, and native plant components that are above the threshold after that second block. The CCD blocking strategy of the PAX is summarized in Figure 2.

The evaluation of the efficiency of this CCD-blocking strategy is described in Section 2.4.4. below.

#### 2.2.4. Positivity Threshold Determination

We determined the PAX positivity cutoff values for the PAX by two methods:

We first tested the sera of nine adult beagles before they were sensitized to Der f in a prior study (courtesy of Prof. Claude Favrot, VetSuisse Faculty, University of Zurich, Switzerland). We then calculated the mean + three standard deviations of all 2160 sIgE values after omitting those specific for the three tropomyosins (Der p 10, Blo t 10, and Per a 7). Indeed, we had confirmed in other studies that IgE against mite tropomyosins is cross-reactive with that of the *Toxocara canis* nematode and is, thus, positive in many healthy dogs, including these laboratory beagles [29].

We then applied the same calculation to the negative control spot (human IgE, which is not recognized by the 5.91 monoclonal antibody) from the cartridges used to test the first cohort of 1019 allergy-suspected dogs during the validation period in 2022.

### 2.3. Validation of the PAX

The studies for validation of the PAX were derived from the following guidelines: EN ISO 13612 [30], CLSI EP17-A2 [31], and I/LA20 [32].

#### 2.3.1. Verification of the Specificity of the Anti-IgE Monoclonal Antibody

Even though the anti-dog IgE mAb 5.91 had been reported not to recognize other canine immunoglobulin isotypes [27], we verified its specificity by ELISA and on the PAX.

Briefly, for ELISA, canine IgE (Bethyl Laboratories, Montgomery, TX, USA) was coated in triplicates on 96-well plates (Maxisorp NUNC, ThermoScientific, Waltham, MA, USA) at 12 decreasing concentrations varying from 200 to 2.31 ng/mL. Dog IgG (Rockland Immunochemicals, Inc., Philadelphia, PA, USA) was coated in triplicates with a ten-fold higher concentration compared with that of IgE. The coated plates were blocked with TBS/0.5% sucrose/0.5% PVP-10 for 1 h at room temperature. They were then washed with TBS/0.05% Tween 20 before incubation with AP-labelled 5.91 at 1 µg/mL for 1 h at room temperature (Dr. Hammerberg, NC State University, Raleigh, NC, USA). After additional washing, the pNPP chromogen (Moss, Inc., Pasadena, MD, USA) was added, and the reading was performed after 30 min.

#### 2.3.2. Verification of the Specificity of IgE Binding to Target Allergens

To demonstrate that irrelevant, perhaps nonpathogenic nematode-specific IgE did not influence the binding of sIgE to their respective allergens on the PAX, we tested ten archived sera from allergy-suspected dogs on the PAX before and after adding 1 µg of monoclonal dog IgE specific for a filarial antigen not present on the PAX (2.39 monoclonal IgE, Bruce Hammerberg, NC State University, Raleigh, NC, USA) [33,34]. The recovery percentage (after-spike values over those before) and coefficient of variation (CV%; standard deviation over the mean) were then calculated for all allergens with sIgE at a concentration above the threshold of 28.00 ng/mL.

#### 2.3.3. Assay Precision

We calculated the precision of the PAX in several ways:

During the validation phase in 2022, we first evaluated the lot-to-lot variability by looking at 54 positive allergen-sample combinations in two different cartridge batches. We also assessed the test’s repeatability by comparing the results of three samples tested in duplicates in five successive test runs. In both cases, we calculated the CV%, as above, for two groups of positivity results: those of Class 1 and 2 (i.e., sIgE < 400 ng/mL) and those of Class 3 and 4 (i.e., ≥ 400 ng/mL).

Finally, to determine the PAX’s reproducibility in a “real-life” testing laboratory situation, we calculated the inter-assay CV% for a positive control pool tested every week between February and December 2023 in one of the European PAX laboratories.

#### 2.3.4. Internal Consistency

We evaluated PAX’s internal consistency by verifying the predicted positive correlation between the sIgE values of two different allergens known to cross-react. Conversely, the sIgE values of two allergens with low or no cross-reactivity are not expected to be correlated.

Among the 23,858 European allergy-suspected dogs tested in 2023, we first selected the sera that had a positive IgE detection (28.00 ng/mL or higher) to either Der f 2 or Der p 2, as these allergens from closely related mite species are predicted to strongly cross-react with an A-RISC coefficient of 0.91 [25]. We calculated the nonparametric Spearman rank correlation coefficient between their corresponding IgE levels. We then repeated the same calculation between the levels of IgE specific for Der f 2 and Tyr p 2 (from *Tyrophagus putrescentiae*), as these two allergens are expected to have little cross-reactivity, even though they belong to the same NPC2 family [25]. Finally, we performed the same calculation between two completely independent allergens, Der f 2 and Par j 2, the nonspecific lipid-transfer protein of the *Parietaria judaica* weed.

#### 2.3.5. Test Performance Characteristics (Sensitivity, Specificity, Accuracy, Predictive Values)

In dogs, the assessment of the sensitivity and specificity of an IgE sensitization test, be it serological or intracutaneous, is intrinsically difficult due to the presence of detectable nonpathogenic IgE sensitizations in both normal and allergic canines. Furthermore, in real-life situations, it is nearly impossible to precisely attribute the flares of clinical signs in an allergic dog to a specific source of allergens in the pet’s environment.

To determine the PAX’s accuracy, we thus obtained sera from two groups of laboratory dogs, which had been archived following the completion of previous research studies. The first group consisted of 11 beagles (courtesy of Prof. Claude Favrot, University of Zurich, Switzerland, and Prof. Wolfgang Bäumer, Free University of Berlin, Germany) and 20 Maltese-beagle crossbred dogs (courtesy of NC State University) sensitized by repeated epicutaneous applications of Der f house dust mites. As these dogs had high levels of Der f-specific IgE after mite sensitization and had experienced skin lesions at the site of Der f application, they were thus proven to have a clinically relevant IgE-mediated allergy to this mite. The control group included 83 non-sensitized beagle dogs deemed healthy and nonallergic by their laboratory animal veterinarian. We restricted our analysis of the PAX metrics to the IgE levels against the Der f house dust mite extract, the only allergen to which the first group was proven allergic and to which the control dogs of the second group were not sensitized.

### 2.4. Influence of Possible Confounding Factors

#### 2.4.1. Effect of Sample Type

To assess whether submitting plasma samples instead of serum (the standard type for IgE serology) would be acceptable, we collected blood by single venipuncture from four atopic dogs. We divided the sample between dry (for serum) and citrate-, heparin-, and EDTA-containing tubes. All tubes were centrifuged to enable the collection of serum and plasma, which were then placed in different tubes and shipped to the laboratory; samples were PAX-tested on the same machine run.

We calculated the recovery percentage and coefficient of variation (CV%) for all allergens with detectable sIgE of at least 28.00 ng/mL and matched results from each of the three types of plasma samples with those of the serum standard. Additionally, we compared these positive values using nonparametric, repeated-measure ANOVA with post-tests to evaluate each plasma type versus serum.

#### 2.4.2. Effect of Hemoglobin, Triglycerides, and Bilirubin

We then wished to evaluate the effect of hemolysis, lipemia, and icterus on sIgE values obtained with the PAX. One dog serum with elevated allergen-specific IgE was spiked with either 15 g/dL (9.3 mmol/L) of hemoglobin, 150 mg/dL (1.69 mmol/L) of triglycerides, or 2.5 mg/dL (42.75 µmol/L) of bilirubin, values corresponding to the serum of a dog with visible hemolytic anemia, hyperlipidemia, or jaundice, respectively. As above, we calculated the recovery percentages and CV% after each type of spike compared with the untreated serum. We also compared all sIgE obtained after each spike to those in the serum using repeated-measure ANOVA with post-tests.

#### 2.4.3. Effect of Sample Storage at Elevated Temperatures

To determine the stability of IgE during shipment to one of the PAX laboratories, we tested five sera from allergy-suspected dogs on Days 0, 3, 7, 14, 21, and 28 after keeping the sera either at room temperature (22 °C on average) or at 37 °C in an incubator, thus simulating a variable transit time in the summer in northern or southern Europe. As above, we calculated the recovery percentage and CV% and compared the positive values using repeated-measure ANOVA, comparing the results obtained at each timepoint to those on Day 0.

#### 2.4.4. Effect of the CCD-IgE Blocking Strategy

We conducted a study to evaluate the effectiveness of the CCD IgE blocking-and-detection method described in Section 2.2.3 above. We chose 16 serum samples from dogs suspected of having allergies, which showed IgE levels above 100 ng/mL against one or both “CCD detectors” in a routine PAX. These samples were then tested three times in the same run: (1) using a dilution buffer without the routine CCD-expressing blocking protein mix, (2) a standard PAX with the first CCD block, and 3) a retest with the second CCD block.

From these three tests, we kept only the IgE specific for pollen, plant food extracts, and native (i.e., CCD-expressing) components. We then calculated the number of positives (IgE ≥ 28.00 ng/mL), the average value of positive IgE, and the percentage reduction in these values after the first and second blocks.

### 2.5. Sensitization of European Dogs to Environmental Allergens

We gathered the results of all 23,858 dogs submitted to one of the three European PAX laboratories (Spain, The Netherlands, United Kingdom). We then calculated the percentage of dogs with at least one positive (specific IgE ≥ 28.00 ng/mL). Also, we determined the average and range of the number of positive sIgE in these dogs and those of positive sIgE levels. We then proceeded to rank the environmental allergens sensitizing these dogs and described in greater detail the sensitization rates to the *Dermatophagoides* house dust mites and their components.

Finally, to confirm the advantage of testing sera with molecular allergens compared with whole crude extracts, we assembled all sIgE values against the Der f 2 mite allergen that were above 28.00 ng/mL and matched them to those against the respective Der f crude extracts in these dogs; values were compared using the Wilcoxon nonparametric paired *t*-test.

### 2.6. Statistics

We used the Wilcoxon matched-pairs signed rank test to analyze the results of the influence of nonspecific IgE, the interference of hemoglobin, triglycerides, and bilirubin, and the association between Der f 2 and Der f values. When comparing the samples collected in plasma and those in serum, as well as when looking at sera stored at room or elevated temperatures, and when evaluating the effect of CCD blocking, we used a nonparametric repeated-measures ANOVA (Friedman test) with post-tests. Finally, to assess the internal consistency of the PAX (i.e., when comparing results between known cross-reactive or non-cross-reactive allergens), we calculated Spearman’s correlation coefficients. The software used was Prism 10.2.2 for Mac (GraphPad, San Diego, CA, USA). All tests were two-tailed, and the threshold for significance was set at 5% (*p* < 0.05).

## 3. Results

### 3.1. Positivity Threshold Determination and Positivity Classes

#### 3.1.1. Positivity Threshold Determination

In the nine healthy beagle dogs, the calculated mean plus three standard deviations of the 2160 observed values was 31.13 ng/mL. When the same computation was applied to the 1019 values of the negative control spot, the result was 25.15 ng/mL. Consequently, we established the threshold for positivity at 28.00 ng/mL, a figure representing the rounded mean of the two above values.

#### 3.1.2. Positivity Classes

To facilitate the graphic representation of allergen-specific IgE levels in the PAX, we arbitrarily set up four positivity classes for allergen-specific IgE as follows:Class 1: 28.00–99.99 ng/mL;Class 2: 100.00–399.99 ng/mL;Class 3: 400.00–799.99 ng/mL;Class 4: ≥800.00 ng/mL.

### 3.2. Validation of the PAX

#### 3.2.1. Verification of the Specificity of the Anti-IgE Monoclonal Antibody

While the 5.91 monoclonal IgE recognized dog IgE with a slope beginning between 59 and 39 ng/mL (serial dilutions 3 and 4), there was no recognition of dog IgG (Figure 3). Altogether, these experiments confirmed that the 5.91 monoclonal antibody recognizes canine IgE but not IgG.

#### 3.2.2. Verification of the Specificity of IgE Binding to Target Allergens

We then wanted to determine if adding a high amount of IgE specific for a nematode allergen not included on the PAX would affect the levels of IgE specific for other allergens spotted on the cartridges.

In the ten canine sera tested, there were 125 detectable allergen-specific IgE of Class 1 positivity, 28 of Class 2, nine of Class 3, and 25 of Class 4. The mean (95% confidence interval; 95% CI) recovery percentages and CV% after the addition of 1 µg of filarial IgE are added to Table 1 below.

Altogether, these 187 sIgE value pairs were not significantly different after the addition of 1 µg of filarial IgE compared with without it (Wilcoxon test, *p* = 0.1546).

These results indicate that the presence of an irrelevant IgE, even at a high amount, does not affect the binding of sIgE to its target.

#### 3.2.3. Assay Precision

During the PAX validation, the intra-assay CV% within each of the two batches were 3.0% and 2.0% for positive results of Class 1–2 and 3–4, respectively. The inter-assay CV% between them was 7.1% and 5.2% for these two different class categories.

Similarly, the intra-assay CV% of sera tested in duplicates in five successive runs were 6.2% and 2.7% for positive results of Class 1–2 and 3–4, respectively. The inter-assay CV% was 8.2% and 7.0% for these two different class pairs.

Finally, in a real-life situation, a single positive control pool yielded 78 positive results, all but one of them being of Class 1 or 2 positivity during the first test. This pool was assayed 56 times using cartridges from ten different batches over 304 days in 2023 in a single laboratory. Altogether, the mean (95% confidence interval [95% CI]) of the coefficient of variation (CV%) for all positive allergens was 12.7% (10.7–14.8).

#### 3.2.4. Internal Consistency

As would be expected between two strongly cross-reactive allergens, the IgE values specific for Der f 2 and Der p 2 were highly significantly correlated (Spearman r = 0.92; *p* < 0.0001; 1322 dogs; Figure 4a). Even though Der f 2 and Tyr p 2 are mite-group 2 allergens, they are expected not to cross-react much, and the correlation between their sIgE was significant but negative (Spearman r = −0.28; *p* < 0.0001; 5032 dogs; Figure 4b); this negative correlation indicates that dogs were generally sensitized to either one of these two allergens, but rarely to both. There was an even stronger negative correlation between levels of IgE to Der f 2 and Par j 2 (Spearman r = −0.49; *p* < 0.0001; 3249 dogs; Figure 4c). The latter two allergens are expected not to cross-react as they belong to different allergen sources (a mite versus a weed) and families (NPC2 versus nonspecific lipid-transfer proteins).

The results above confirm that the PAX has excellent internal consistency; it finds relationships between related allergens but few concurrent positives between unrelated allergen pairs.

#### 3.2.5. Test Performance Characteristics

The allergic group included 31 dogs, experimentally sensitized and clinically reacting to Der f, while the healthy group consisted of 83 Der f-non-sensitized beagles. Dogs were considered sensitized to Der f if they had sIgE levels of 28.00 ng/mL or higher to this mite extract; their sensitization status can be found in Table 2 below.

All but four mite-allergic dogs had a positive PAX test to the Der f extract (IgE ≥ 28.00 ng/mL), while all but two of the mite-non-sensitized dogs had IgE values lower than 28.00 ng/mL.

From this 2 × 2 table, we calculated the following parameters:Sensitivity = 27/(27 + 4) = 87.1%;Specificity = 81/(81 + 2) = 97.6%;Accuracy = (27 + 81)/(27 + 2+4 + 81) = 94.7%;Positive predictive value = 27/(27 + 2) = 93.1%;Negative predictive value = 81/(81 + 4) = 95.3%.

Overall, these favorable performance metrics confirmed the high accuracy of the PAX.

### 3.3. Influence of Possible Confounding Factors

#### 3.3.1. Effect of Sample Type

In these four allergy-suspected dogs, there were 25 allergen-specific IgE values of at least 28.00 ng/mL: 15 (60%) were of positivity Class 1, three (12%) of Class 2, four (16%) of Class 3, and two (8%) of Class 4.

Table 3 includes the means (95% CI) of the recovery percentages and CV% for all three types of plasma compared with serum.

Overall, some changes were seen when using plasma instead of serum: several Class 1 positives in serum became negative with plasma (three in citrate-, four in heparin-, and five in EDTA-collected samples), and there were also increases or decreases of one positivity class for some IgE values (one with citrate, two with heparin, and two with EDTA).

The sIgE values with the three types of plasma samples were significantly different from their respective ones in the serum (Friedman repeated-measures ANOVA, *p* = 0.0029). More specifically, the values of citrate-collected samples were significantly different from those in the serum (Dunn’s multiple comparison tests: *p* = 0.0014), while those taken in heparin and EDTA were not (*p* = 0.9725 and 0.0643, respectively).

Altogether, because of the negativization of some low-positive sIgE with each type of plasma sample, it is preferable to submit serum rather than plasma for PAX testing.

#### 3.3.2. Effect of Sample Storage at Elevated Temperatures

To simulate a long shipping time to the laboratory, we repeatedly PAX-tested five canine sera stored at room temperature (around +22 °C) or +37 °C for 28 days.

On the first day of testing (Day 0), 32 positive allergen-specific IgE results were obtained: 27 (84%) of Class 1, four (13%) of Class 2, and two (6%) of Class 3.

Table 4 and Table 5 include the recovery percentages and CV% of the sIgE values when serum was stored at room temperature or +37 °C for 28 days, respectively.

The storage of serum at room temperature (around +22 °C) for 28 days had some minor impact on the 32 sIgE values already positive on Day 0. The lowest recovery percentages and highest CV% were all seen in samples from one of the five dogs (Dog 3) whose values began declining as soon as Day 3; one of these sIgE values eventually experienced a drop of over 50% to become negative (26.43 ng/mL) on Days 21 and 28.

Still, overall, the repeated testing of the five sera kept at room temperature six times over 28 days did not yield significantly different results (one-way repeated-measure ANOVA; *p* = 0.1436), with the IgE values at none of the later days being significantly different from those on Day 0.

When the sera were kept at the higher temperature of +37 °C, the sIgE values did not change much for two of the five dogs. For the three others (Dogs 2, 3, and 5), however, several sIgE values had decreased to only ~30–60% of the original ones on Day 28, with one of the 32 IgE values having negativized to 27.07 ng/mL.

Despite the observations above, sIgE values were not significantly different over the 28 days of testing (one-way repeated-measure ANOVA; *p* = 0.099), and the later tested values were not significantly different from those on Day 0.

#### 3.3.3. Effect of Hemoglobin, Triglycerides, and Bilirubin

We tested the serum of one dog before and after spiking it with a high amount of either hemoglobin, triglycerides, or bilirubin.

In this dog, there were 26 positive sIgE values: 13 (50%) were of positivity Class 1, seven (27%) of Class 2, four (15%) of Class 3, and two (8%) of Class 4.

The recovery percentages and coefficient of variation after spiking with these three compounds are included in Table 6.

After spiking with triglycerides or bilirubin, all sIgE were within 25% of their original values. The highest variation was seen after hemoglobin spiking, with three very low Class 1 positive IgE values negativizing with a loss of more than 25% of their original levels.

Despite this observation, the sIgE levels were not significantly different after spiking with hemoglobin, triglycerides, or bilirubin compared with without such an interference (Wilcoxon; *p* = 0.1227; 0.4525, and 0.1105, respectively).

Overall, these results suggest that hemolysis, lipemia, or icterus have little influence on PAX-determined sIgE values. However, severe hemolysis may affect the detection of low sIgE levels.

#### 3.3.4. Effect of CCD-IgE Blocking Strategy

In PAX version 22.2, 55 allergens from pollens and plant foods harbored CCDs (44 extracts, 11 components), and two additional CCD-expressing “detectors” were also included.

When the 16 selected canine sera were tested on the PAX in a dilution buffer without the routine CCD-IgE blockers, the mean (95% CI) number of positive sIgE (i.e., ≥28.00 ng/mL) against CCD-expressing allergens was 26.1 (18.5–33.7), and the mean (95% CI) level of these sIgE was 122.7 ng/mL (61.0–184.5). The percentage reduction in these values after the first routine and second CCD blocks is shown in Table 7. The first block removed, on average, slightly more than half of the positives and one-third of the sIgE values against CCD-expressing allergens. The second inhibited about three-fourths of these positives and half of the sIgE values against these allergens.

The sIgE values were significantly different after the first and second blocks compared with those without blocking (Friedmann test, *p* < 0.0001 overall, and after each block compared with without).

The combination of both CCD-IgE blocks negativized (i.e., sIgE < 28.00 ng/mL) one of the “CCD detectors” in three dogs (18.8%) and both in 7/16 dogs (43.8%). This CCD-IgE blocking strategy negativized the sIgE to all CCD-expressing allergen extracts and components in three dogs (18.8%).

### 3.4. Sensitization of European Dogs to Environmental Allergens

The PAX testing of 23,858 dogs in Europe yielded, after removing the values below the detection limit, 3,030,375 interpretable individual sIgE levels against environmental and insect venom allergens. Among these were 91,489 separate sIgE values (3.0%) equal to or greater than 28.00 ng/mL. These allergen-specific values were distributed among the four positivity classes as follows:Class 1: 73,947 (80.8%);Class 2: 12,532 (13.7%);Class 3: 2427 (2.7%);Class 4: 2583 (2.8%).

The range of sIgE detection was 10.24 to 1826.70 ng/mL. The average (95% CI) of all sIgE values against environmental and venom allergens above the positive threshold was 129.21 ng/mL (128.66–129.76).

Altogether, 18,825 dogs (78.9% of those tested) had at least one positive sIgE against environmental or venom allergens, and the mean (95% CI) of positives per dog was 3.8 (3.8–3.9). Finally, there were 17,548 dogs (73.6% of those tested) that had at least one sIgE value against an environmental allergen treatable by immunotherapy, if clinically relevant. In these dogs, the mean (95% CI) number of allergens suitable for immunotherapy was 2.9 (2.8–2.9).

In Appendix A, we have listed the seropositivity rates for all allergens tested in 23,858 dogs, ranked in decreasing order. The 20 most common allergens that sensitized dogs in Europe in 2023 were those of house dust and storage mites, honeybee and wasp venoms, and some weed and tree pollens (Appendix A). In these dogs, the most detected allergen was the honeybee venom’s phospholipase A2, Api m 1 (22.8%). The rarest (0.1% or less) were the epithelial lipocalins from dogs (Can f 2 and Can f 6) and mice (Mus m 1), a not surprising finding as dogs are unlikely to sensitize to their own proteins.

The rates of IgE seropositivity to *Dermatophagoides* allergens were, in decreasing order, the following: the Der f extract (60.8%), the Der p extract (5.9%), Der f 2 (5.4%), Der f 1 (4.9%), Der p 2 (4.7%), Der p 1 (2.6%), Der p 11 (2.6%), Der p 10 and Der p 21 (2.4%), Der p 5 (1.8%), Der p 20 (1.2%), Der p 23 (1.1%), Der f 15 and Der f 18 (0.6%), and Der p 7 (0.4%).

Finally, we selected all dogs for which the Der f 2 sIgE serum level was at least 28.00 ng/mL and paired each dog’s Der f 2 serum level with that against its parental Der f extract (Figure 5).

There were 1283/23,858 dogs (5.4%) that had sIgE against Der f 2 above the positivity threshold. In 1000 of these dogs (77.9%), the sIgE values against Der f 2 were higher than those against the Der f extract, with the mean (95% CI) IgE levels against the component (401.6 ng/mL [375.0–428.3]) being about three times higher than that against the extract (126.5 ng/mL [115.2–137.8]). In fact, in 380/1283 dogs (29.6%), the sIgE level against the Der f extract was less than 28.00 ng/mL (i.e., the Der f sIgE would be negative in the PAX) while that against Der f 2 was higher than the positive threshold in the same dog. Altogether, the Der f 2 sIgE values were significantly higher than those against Der f (Wilcoxon paired *t*-test, *p* < 0.0001), thereby confirming the interest in testing against a molecular allergen rather than with its parent extract that might have a low content of that allergen.

Of interest is that in 283/1283 dogs (22.0%), the Der f-specific IgE were higher than those against Der f 2, suggesting that these dogs’ sensitization to Der f likely was directed against (a) mite component(s) other than Der f 2.

## 4. Discussion

In this paper, we presented the development and validation of the first quantitative multiplex macroarray specifically designed for the determination of IgE sensitizations in dogs. This PAX platform is the first to enable the determination of sIgE against both molecular allergens and allergen extracts in this species. Additionally, we report herein the IgE seropositivity rate to environmental and insect venom allergens in nearly 24,000 dogs tested in Europe in 2023; this represents the largest sensitization dataset reported to date in veterinary allergology.

### 4.1. PAX Design

#### 4.1.1. Allergen Selection

The PAX, like its ALEX^2^ human counterpart, currently includes around one-third of allergen extracts and two-thirds of separate molecular allergen spots. More specifically, the version of the PAX discussed in this paper contains 247 allergen spots, with 145 (58.7%) containing environmental or *Hymenoptera* insect venom allergens. Excluding the “CCD detectors,” the remaining 98 spots (39.7%) are coated with food allergens, the results of which will be discussed in another paper. At the time of this writing, the PAX stands as the largest commercially available serological test for detecting IgE sensitivities in dogs.

Although there exists comprehensive knowledge on the identification and clinical relevance of molecular allergens in human allergic patients, as demonstrated in the second User’s Guide of Molecular Allergology [11], there have been only a few reports of molecular allergens identified by IgE in allergic dogs, with the majority being food allergens (refer to Section 2.2.1 above). Therefore, our initial selection of molecular allergens for the PAX was based on the hypothesis that proteins that trigger IgE responses in humans are likely to elicit similar reactions in animals. This hypothesis was confirmed, as all components included in the PAX were found to be IgE-targeted in multiple dogs (Appendix A). Our results also confirmed that previously characterized allergens for dogs were, indeed, allergenic in this species. The wide array of allergens tested in the PAX thus allows for a notable expansion of the canine allergome. There is a high likelihood that there are additional prevalent and relevant molecular components in allergic dogs that remain unidentified. Current studies are underway to pinpoint these molecular allergens for their potential inclusion in upcoming PAX versions.

#### 4.1.2. IgE Capture Reagent

In the PAX for dogs, we selected the well-characterized anti-IgE monoclonal antibody 5.91 developed by Prof. Bruce Hammerberg at NC State University. This antibody targets a unique epitope in the CH2 domain of the dog’s epsilon chain, which has very little similarity with the sequences of the four dog IgG subclasses A to D, IgM, and IgA. While the 5.91 had already been reported not to recognize canine IgG, IgA, and IgM [27], we confirmed using ELISA that it, indeed, did not bind to dog IgG.

#### 4.1.3. Positivity Threshold Determination

Determining the positivity threshold for an IgE serological test in allergic dogs is challenging, as nonpathogenic (clinically irrelevant) allergen-specific IgE is commonly found in healthy dogs, and some dogs with clinical signs of allergy have negative intracutaneous and serological sensitization tests (e.g., those diagnosed with “atopic-like dermatitis”) [35]. As a result, it is impossible to select a discriminating cutoff using a receiver operating characteristic (ROC) curve due to the lack of clearly separated populations with expected positive/negative serological test results in this species.

Thus, for the PAX, we used the traditional method of setting the positivity threshold as the mean plus three standard deviations of more than 2000 sIgE values obtained after testing nine healthy beagle dogs. We then repeated the same calculation on more than 1000 replicates of the negative control (human IgE) on the PAX; the chosen threshold of 28.00 ng/mL was the rounded mean of these two calculations.

Setting up a “positivity threshold” does not exclude the presence of pathogenic and clinically relevant sIgE at levels below the chosen cutoff value. Indeed, the pathogenicity of sIgE extends beyond its serum level, as one should also consider its affinity to target allergens or to the high-affinity IgE receptor, its glycosylation status, and its conformation. The rationale of a simple, dichotomous, “negative/positive” interpretation of IgE serological test results based on fixed positivity thresholds is being questioned in human allergology [36]. As for other tests, PAX results must always be interpreted in conjunction with the patient’s medical history and clinical symptomatology.

### 4.2. PAX Validation

In the interest of brevity, this section will focus on just four key aspects of the PAX validation.

#### 4.2.1. Verification of the Specificity of IgE Binding to Target Allergens

A valid concern for any serological assay used for the determination of sensitizations is whether the detection of allergen-specific IgE is affected by clinically irrelevant IgE (e.g., nematode-specific IgE, which is often present at high levels in normal and allergic dogs) [37]. As reported in Section 3.2.2 above, spiking ten sera with 1 µg of a monoclonal IgE specific for a filarial antigen had almost no influence on the detection of 200 sIgE of varying positivity classes. These observations suggest that high levels of irrelevant IgE are not likely to interfere with the binding of an IgE to its specific allergen on the PAX.

#### 4.2.2. Assay Precision

In Section 3.2.3, we discussed the PAX’s batch-to-batch variability and repeatability (intra- and inter-assay coefficients of variation). These were evaluated during the validation process and over its first year of use in a single laboratory. In both situations, the PAX demonstrated very good reproducibility, which was nearly identical to that of other ELISA-based allergen-specific IgE serological assays for dogs at the time of development [38] or during one year of use [39].

#### 4.2.3. Internal Consistency

As a measurement of the internal consistency of the PAX, we wanted to see if the levels of IgE specific for two allergens that are known to cross-react would show a positive correlation; in contrast, those not expected to cross-react would not be so correlated. Our findings support the PAX’s excellent internal consistency, as we observed a strong positive correlation between the two *Dermatophagoides* NPC2 proteins but a negative one between Der f 2 and Tyr p 2, as well as between Der f 2 and the unrelated *Parietaria* nonspecific lipid-transfer protein Par j 2; these results perfectly align with the theoretical prediction of cross-reactivity between these allergens [25].

#### 4.2.4. Test Performance Characteristics

As mentioned above in Section 2.3.5 and Section 4.1.3, determining the accuracy of any serological or intracutaneous sensitization test is inherently difficult due to the presence of clinically irrelevant IgE in healthy dogs and allergic dogs without detectable sensitizations. Furthermore, precisely identifying the allergenic cause of a flare will be nearly impossible for environmental allergens. Therefore, we decided to evaluate the PAX’s accuracy in laboratory dogs with or without experimental sensitization to the Der f house dust mite. All sensitized dogs had experienced flares of skin lesions after epicutaneous application of the Der f extract, indicating mite allergy. Almost all allergic dogs had a positive PAX test to the Der f extract, while nearly all non-sensitized beagles had a negative test. Based on these results, the PAX demonstrated a sensitivity near 90% and a specificity above 97%. This very high accuracy of the PAX cannot be compared with other veterinary IgE serological tests, as similar data have not been reported previously for these assays.

In dogs with unambiguous *Hymenoptera* venom-induced anaphylaxis in dogs, the PAX also displayed high sensitivity to detect bee or wasp venom-specific IgE [40]. Conversely, our unpublished data revealed that only one out of 77 healthy beagles living indoors (1.2%) had IgE against such allergens, an excellent specificity. Altogether, these results confirm the high accuracy of the PAX in the case of unequivocal natural allergy.

### 4.3. Influence of Possible Confounding Factors

Several situations could potentially affect the PAX’s performance: the type of samples used, high temperatures during the shipment of sera to the laboratory in summer months, hemolysis, lipemia, icterus, and the presence of CCD-specific IgE; separate investigations were performed to evaluate each of these potential confounding factors.

In one experiment, PAX results were compared between plasma and serum samples. These indicated that some low-positive sIgE values in serum could become negative when submitted as plasma. As a result, it is advisable to submit only serum samples, not plasma, to ensure accurate results in the PAX.

We then stored five sera at room temperature (around 22 °C) or at 37 °C and tested them once weekly for four weeks. The storage at room temperature for four weeks had minimal impact on the sIgE values for four of five dogs, while those of the fifth had lower testing reproducibility; the duration and temperature during shipment of that serum prior to arriving at the laboratory regrettably were not available. The variability of PAX results was higher when the sera were stored at 37 °C for longer than two weeks. These observations suggest that while sIgE levels remain somewhat stable over time, the longer sera are stored at high temperatures, the higher the variability of PAX results. These findings are consistent with recent reports on the sIgE ELISA testing of canine sera stored at elevated temperatures [41,42]. Consequently, it is advisable to ensure that the transit duration of the samples to the laboratory is two weeks or less, especially in the summer and in warmer climates.

In other experiments, the serum of a dog suspected of allergy was spiked with either hemoglobin, triglycerides, or bilirubin to simulate full hemolysis, lipemia, or icterus, respectively. Despite PAX results not being significantly different with or without a spike, it is noteworthy that some low sIgE values could become negative in the case of high hemoglobinemia, an important factor to consider.

As CCD-specific IgE is known to cause false positive serological tests to pollen and plant food extracts [43], it is essential to eliminate such interference. As described in Section 2.2.3 above, the strategy used for CCD blocking in the ALEX^2^ was expanded in the PAX. Compared other IgE serological assays for veterinary use, a unique characteristic of the PAX is its inclusion of two “CCD detectors,” which are human proteins harboring different types of CCDs. These “CCD detectors” allow for a verification of the efficiency of the CCD blocking.

In this paper, we reported the results of the evaluation of the PAX CCD blocking strategy in 16 dogs (Section 3.3.4). The routine block with a proprietary mix of CCD-expressing proteins permitted the removal of about one-third of sIgE against CCD-harboring allergens; the second block, half of these. Moreover, the CCD-blockers markedly reduced the number of positive sIgE against CCD-expressing allergens: half after the first block and three-fourths after the second. Of importance is that any persistence of elevated sIgE levels against “CCD detectors” after the two blocks would identify dogs in which the CCD-blocking was insufficient, and veterinarians would be made aware of the possible clinical irrelevance of sIgE against pollen extracts and native pollen components.

Based on our current knowledge, other commercial ELISA-based serological tests for veterinary use do not utilize “CCD detectors” to assess the effectiveness of CCD blockers. Our use of these detectors allowed us to identify dogs with very high levels of anti-CCD IgE. We also found cases where two rounds of CCD blocking did not completely eliminate such IgE. Without CCD detectors to identify these cases, serological results may show multiple sensitizations to pollen extracts, which could lead to unnecessary immunotherapy in dogs that are unlikely to benefit from it.

### 4.4. Sensitization of European Dogs to Environmental Allergens

#### 4.4.1. Environmental and Venom Allergens

In 2023, nearly 24,000 dogs suspected of having allergies were tested in Europe with the PAX. The testing yielded over three million values of IgE specific for environmental and insect venom allergens. Although the percentage of sIgE above the chosen positivity threshold may seem low (3.0%), it is important to note that between one and ten allergen spots in the PAX could correspond to a single allergen source. Therefore, not being sensitized to that specific allergen source could result in up to ten sIgE values falling below the chosen threshold. Moreover, the relatively low number of positive sIgE, especially against pollens, indicates the effectiveness of the CCD-blocking strategy used in the test.

In these dogs, most of the positive sIgE detected were of Class 1 (28.00 to 99.99 ng/mL). However, the level of sIgE does not necessarily correlate with the severity of clinical signs of allergy. Indeed, a study of dogs with *Hymenoptera* venom anaphylaxis revealed that all but one dog with severe signs had only Class 1 positive sIgE to honeybee venom, and the only dog with mild signs had high Class 2 sIgE to that extract [40].

When looking at the allergens most commonly sensitizing these dogs, three groups of allergens dominated: extracts and components of allergens from storage and house dust mites, *Hymenoptera* venoms, and some pollens.

The most commonIgE sensitization was to the Der f extract, as would be expected due to its cross-reactivity with the ubiquitous nematode allergen *Toxocara canis* [44].

Our PAX testing results revealed, for the first time, that about 20% of dogs were stung at least once by honeybees or wasps, which led to their sensitization to venom extracts and components; this sensitization rate appears to be half that of humans [45]. Of importance is that the presence of long-lasting serum sIgE in insect venoms does not always herald the presence of clinical allergy to bee or wasp stings [45]. Our results also establish that Api m 1, Api m 3, and Api m 10 are major allergens from the honeybee venom, as in humans in whom sensitization to these allergens indicates a primary sensitization to that insect’s venom [45]. Similarly, the main wasp allergens for dogs are Ves v 5 and Ves v 1, which in humans indicate a primary sensitization to wasps [46].

In the PAX, the most detected pollen sensitizations were those to extracts from three weeds: ragweed (*Ambrosia artemisiifolia*; Amb a), wall pellitory (*Parietaria judaica*; Par j), and the Russian thistle (*Salsola kali*; Sal k). There were nearly identical sensitization rates to the extract of Par j and its nonspecific lipid-transfer protein Par j 2, the marker of primary sensitization to this weed [11].

Also prevalent were sensitizations to the cypress (*Cupressus sempervirens*) pollen extract (Cup s) and the PR-10-family allergen Fag s 1 from the European beech (*Fagus sylvestris*). These sensitizations may reflect the testing of allergic dogs in southern (Cup s) or central continental Europe (Fag s 1). Of interest is that the sensitization to Fag s 1 appeared far more prevalent than that to the other PR-10 pollen allergens in the PAX, Bet v 1 from birch (*Betula verrucosa*), Aln g 1 from alder (*Alnus glutinosa*) or Cor a 1 from hazel (*Corylus avellana*). This difference likely highlights that beech, after years of reforestation, is now the most dominant deciduous tree in continental Europe [47].

Finally, the similarity of sensitization rates seen between related extracts and components (e.g., Api m and Api m 1, Par j and Par j 2) is another marker of the PAX’s excellent internal consistency.

Subsequent papers will thoroughly discuss the seasonal dynamics of pollen sensitizations and the clustering of sensitizations in this group of dogs.

#### 4.4.2. House Dust Mite Allergens

In this group of dogs, the ten-fold difference in the IgE seropositivity rates to Der f and its closely related species, *Dermatophagoides pteronyssinus* (Der p), was a perplexing finding. Unlike Der f, the Der p extract seems not to cross-react with the excretory-secretory extract of Toxocara canis, which could explain the disparity between their respective sensitization rates [29].

The *Dermatophagoides* mite components most commonly sensitizing dogs in this cohort were those of group 2 (Der f 2, Der p 2) and group 1 (Der f 1, Der p 1), components that also most commonly sensitize human patients [48]. Conversely, we detected a surprisingly low rate (0.6%) of sensitizations to the high-molecular-weight mite allergens Der f 15 and Der f 18, which were previously reported to be major *Dermatophagoides* mite allergens for dogs [49,50]. The explanation for this discrepancy lies in our recent discovery that IgE directed against complex glycans from secretory mucins of *Toxocara canis* cross-react with similar carbohydrates on the native high-molecular-weight allergens Der f 15 and Zen-1 (and likely Der f 18) [29]. The Der f 15 and Der f 18 spotted on the PAX do not have natural glycans, and they are thus not detected by these cross-reactive IgE. Other cross-reactive allergens between *Toxocara canis* and *Dermatophagoides* mites are tropomyosins (mite group 10) and paramyosins (mite group 11) [29]. Because of this cross-reactivity with omnipresent ascarid allergens, the clinical relevance of IgE against the glycans of Der f 15, Zen-1 (and likely Der f 18), as well as against Der p 10 and Der p 11, is unclear.

Finally, we selected dogs with elevated sIgE against Der f 2 and matched these concentrations against those to the Der f parent extract. The fact that the IgE levels against the component were, on average, three times higher than against the extract emphasizes the value of molecular allergology. That almost one-third of the dogs with Der f 2-sIgE levels above the positivity threshold also had a corresponding sIgE level against the Der f extract lower than that threshold also supports the benefit of using molecular allergen testing. The latter observations are not surprising, as components such as Der f 2 might only represent a small percentage of the proteins in a Der f extract [10].

## 5. Conclusions

In this paper, we reported the development and validation of the first multiplex macroarray, which includes molecular allergens, for identifying allergen sensitizations in dogs. The PAX, adapted from the ALEX^2^ for humans, is accurate and consistent. It features a unique method for identifying and inhibiting CCD-specific IgE. Molecular allergology provides an exceptional understanding of allergen cross-reactivities, which could help improve immunotherapy formulations for dogs suffering from environmental allergies.

## Figures and Tables

**Figure 1 vetsci-11-00482-f001:**
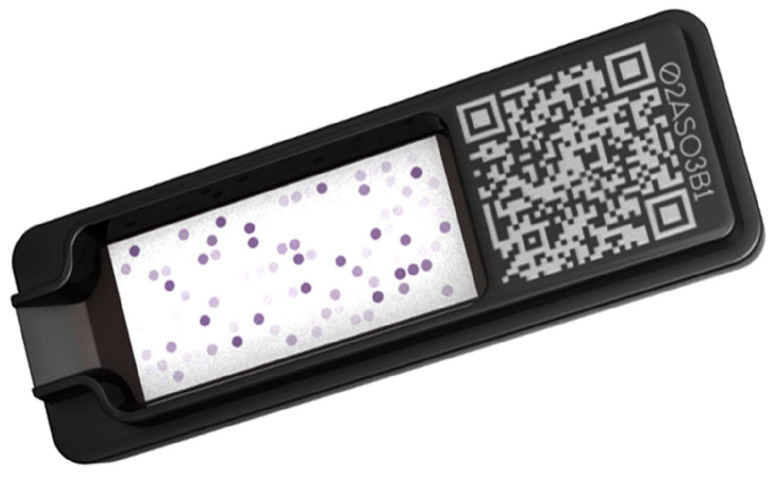
Representation of an ALEX^2^/PAX cartridge, which enables the simultaneous multiplex testing of up to 300 allergens and their controls. Photo courtesy of MacroArray Diagnostics, Vienna, Austria.

**Figure 2 vetsci-11-00482-f002:**
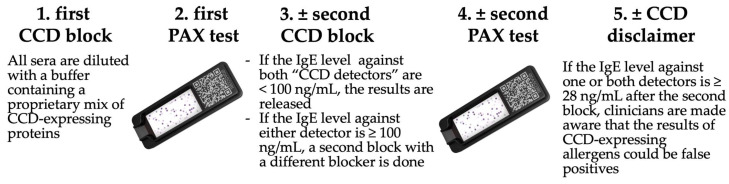
PAX CCD-blocking strategy.

**Figure 3 vetsci-11-00482-f003:**
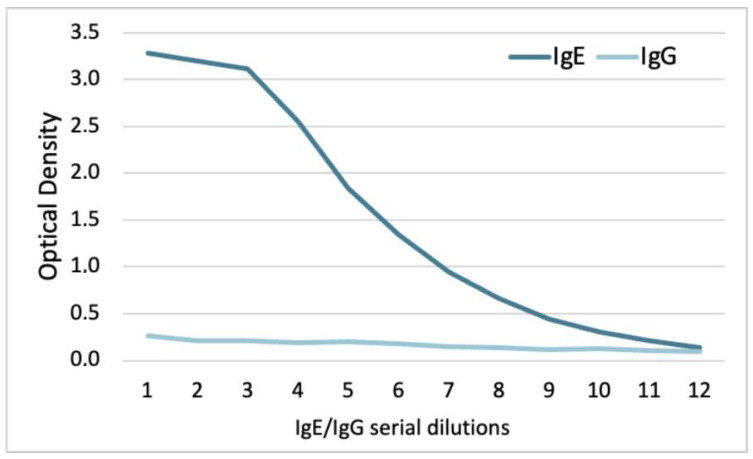
ELISA with the 5.91 anti-dog IgE monoclonal antibody: this antibody recognized dog IgE but not dog IgG.

**Figure 4 vetsci-11-00482-f004:**
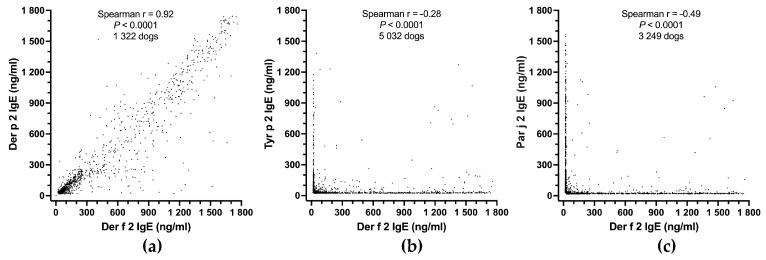
PAX internal consistency: there is a strong positive correlation between the IgE values against the cross-reactive allergens Der f 2 and Der p 2 (**a**), but a negative correlation between those of two non-cross-reactive allergen pairs, Der f 2 and Tyr p 2 (**b**), or Der f 2 and Par j 2 (**c**).

**Figure 5 vetsci-11-00482-f005:**
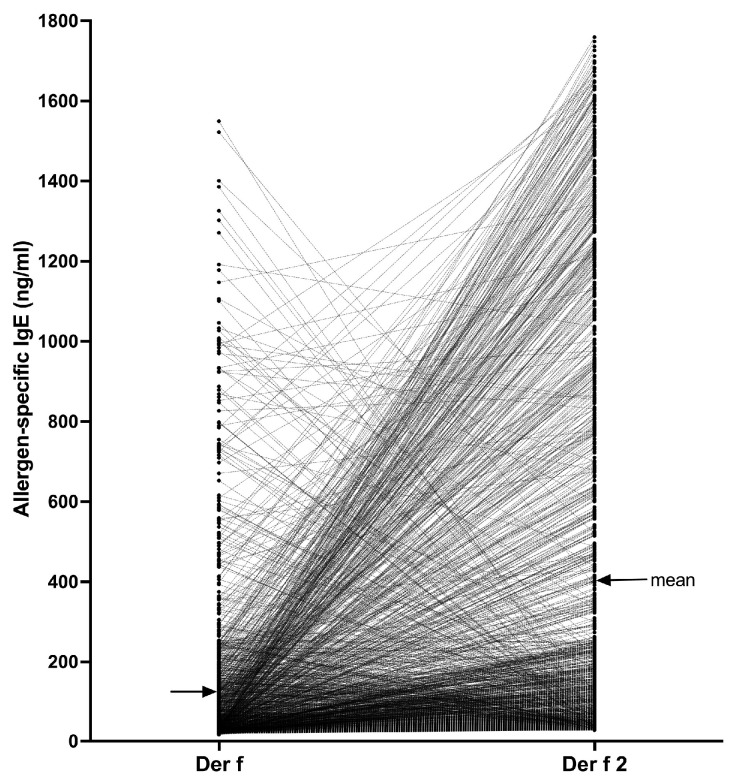
Comparison of Der f 2 and Der f-specific IgE values in 1283 dogs: In most dogs, the IgE levels against Der f 2 exceeded those against the Der f extract.

**Table 1 vetsci-11-00482-t001:** Mean (95% confidence interval) recovery percentages and coefficients of variation (CV%).

	Class 1	Class 2	Class 3	Class 4
**Recovery percentage**	99.5% [98.1–101.0]	99.0% [95.5–102.6]	99.6% [84.1–115.0]	98.9% [96.5–101.3]
**Coefficient of variation**	4.2% [3.5–4.8]	4.4% [2.3–6.4]	9.7% [0.0–19.9]	3.0% [1.8–4.2%]

**Table 2 vetsci-11-00482-t002:** Sensitization status to the Der f house dust mites.

	Allergic Dogs	Healthy Dogs
**Der f sIgE ≥ 28.00 ng/mL**	27	2
**Der f sIgE < 28.00 ng/mL**	4	81

**Table 3 vetsci-11-00482-t003:** Mean (95% confidence interval) recovery percentages and coefficients of variation (CV%).

	Citrate	Heparin	EDTA
**Recovery percentage**	91.3% [88.3–94.2]	100.5% [93.3–107.7]	106.6% [89.4–123.9]
**Coefficient of variation**	6.8% [4.6–9.1]	7.7% [4.5–10.8]	13.3% [7.2–19.4]

**Table 4 vetsci-11-00482-t004:** Mean (95% confidence interval) recovery percentages and coefficients of variation (CV%) of sera stored at room temperature (around +22 °C) for 28 days.

	Day 3	Day 7	Day 14	Day 21	Day 28
**Recovery percentage**	99.7% [95.1–104.2]	104.5% [97.8–111.2]	101.9% [94.5–109.3]	103.1% [95.5–110.7]	98.4% [91.6–105.2]
**Coefficient of variation**	7.0% [4.8–9.3]	10.0% [6.5–13.5]	11.5% [7.9–15.0]	10.9% [6.7–15.1%]	10.4% [6.4–14.3%]

**Table 5 vetsci-11-00482-t005:** Mean (95% confidence interval) recovery percentages and coefficients of variation (CV%) of sera stored at +37 °C for 28 days.

	Day 3	Day 7	Day 14	Day 21	Day 28
**Recovery percentage**	99.0% [92.4–105.7]	98.5% [91.2–105.8]	103.9% [93.7–114.0]	96.6% [86.6–106.5]	96.8% [85.5–108.1]
**Coefficient of variation**	9.7% [6.0–13.4]	4.5% [6.0–15.0]	15.7% [10.9–20.5]	17.0% [11.8–22.2%]	18.9% [12.4–25.4%]

**Table 6 vetsci-11-00482-t006:** Mean (95% confidence interval) recovery percentages and coefficients of variation (CV%).

	Hemolysis	Lipemia	Bilirubinemia
**Recovery percentage**	93.2% [86.3–100.2]	99.1% [96.1–102.0]	96.9% [93.0–100.8]
**Coefficient of variation**	10.7% [6.4–14.9]	4.4% [3.3–5.6]	5.9% [4.1–7.7]

**Table 7 vetsci-11-00482-t007:** Mean (95% confidence interval) percentage change in CCD-specific IgE after the first and second CCD-IgE blocks.

	After the First CCD Block	After the Second CCD Block
**Reduction in positive numbers**	54.9% [41.5–58.4]	72.1% [56.0–88.1]
**Reduction in positive IgE level**	33.5% [16.7–50.4]	50.5% [28.4–72.5]

## Data Availability

The ranking of all seropositivity results can be found in Appendix A.

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
