# Peer review of "Validation of a Multiplex Molecular Macroarray for the Determination of Allergen-Specific IgE Sensitizations in Dogs"

_vetsci, 2024, doi:10.3390/vetsci11100482_

Round 1

Reviewer 1 Report

Comments and Suggestions for Authors

This manuscript by Olivry et al describes a novel multiplex assay to identify allergen-specific IgE in dogs.  The subject matter and material are quite relevant to small animal practice and specifically to canine dermatology.  The manuscript is well written and easy to follow.  There are several revisions needed before this manuscript is acceptable for publication.

Minor revisions:

Lines 721-729: Insect venom sensitivity: was this confirmed by the intradermal skin testing in any of the dogs?  If so this should be mentioned.  If intradermal skin testing was not performed, the limitations of relying upon anti-IgE alone should be added to the discussion.

Major Revisions:

1. More explanation of complex carbohydrate blocking is needed in the methods.  A figure with a diagram might be useful for those unfamiliar.

2. Line 750-752 references unpublished data regarding Toxocara canis cross reactivity with allegen-specific IgE.  This is unacceptable for a methodology paper describing a new method and represent a serious omission.  This reference to unpublished data should be replaced by a supplementary figure and / or a reference to a publicly available source.

Author Response

This manuscript by Olivry et al describes a novel multiplex assay to identify allergen-specific IgE in dogs.  The subject matter and material are quite relevant to small animal practice and specifically to canine dermatology.  The manuscript is well written and easy to follow.  There are several revisions needed before this manuscript is acceptable for publication.

Minor revisions:

Lines 721-729: Insect venom sensitivity: was this confirmed by the intradermal skin testing in any of the dogs?  If so this should be mentioned.  If intradermal skin testing was not performed, the limitations of relying upon anti-IgE alone should be added to the discussion.

 Response: Our dataset used is near 24,000 dogs, and veterinarians do not normally include skin test results in their submissions. We make clear in this paragraph that we are dealing only with the PAX results. Moreover, we disagree that using the PAX alone is a limitation. Indeed, in our reference 40 (Rostaher et al), the PAX had a higher sensitivity than both IDT and prick test for the determination of Hymenoptera venom sensitization in dogs with anaphylaxis to insect stings. This, coupled with its excellent specificity (none of the beagles living indoors are PAX-positive with venom allergens; Nextmune data), make the PAX superior to IDT for venom allergy in dogs. Consequently, there is no reason to add a limitation about venom testing.

Major Revisions:

  1. More explanation of complex carbohydrate blocking is needed in the methods.  A figure with a diagram might be useful for those unfamiliar.

Response: We added a new figure 2 to highlight the CCD blocking process

.

  1. Line 750-752 references unpublished data regarding Toxocara canis cross reactivity with allegen-specific IgE.  This is unacceptable for a methodology paper describing a new method and represents a serious omission.  This reference to unpublished data should be replaced by a supplementary figure and / or a reference to a publicly available source.

Response: We replaced the “unpublished data” with reference number 29, which is of a newly accepted paper.

Reviewer 2 Report

Comments and Suggestions for Authors

General commentary

The manuscript deals with the development and validation of a new diagnostic molecular test for veterinary medicine, which is currently unique in the world. It has been used in three laboratories in Europe over the last two years and tested on more than 23000 dog sera. The topic is original and absolutely relevant to the field as it fills a specific gap in the field. Compared to other publications, the research described in the manuscript provides data for the development and validation of a completely new diagnostic method that cannot be compared to other diagnostic methods in the field of veterinary allergology due to the different, i.e. molecular, methods used. Older methods were all based on the ELISA test. The research is based on the methodology and the number of animals, which is sufficient to support the conclusions. The conclusions are consistent with the evidence and arguments presented and address the main question posed. References are appropriate. The tables and figures are representative.

The manuscript is well written, with a clear structure, perfectly written material and methods and presentation of results, with a structured discussion and clearly understandable conclusions after each part of the discussion. All of us who deal with allergic dogs on a daily basis, as well as researchers in the field of veterinary dermatology, can only learn from this article.

Minor notes:

1.           Reference no. 16= Reference no. 14* and this must be corrected throughout the text, therefore:

Line 80:...« or singleplex molecular assays [13-20].« should be corrected to [13-19]

Consequently, from line 101 to line 232, all numbers for references should be reduced by one number.

The citation numbers between lines 582-729 and 742-765 must also be corrected (reduced by one number)

5.  Lines 182-184: the sentence »Indeed, we had confirmed in other studies that IgE against mite tropomyosins is cross reactive with that of the Toxocara canis nematode and is thus positive in many healthy dogs, including these laboratory beagles.« needs to be referenced.

*Scala, E.; Caprini, E.; Abeni, D.; Meneguzzi, G.; Buzzulini, F.; Cecchi, L.; Villalta, D.; Asero, R. A qualitative and quantitative comparison of IgE antibody profiles with two multiplex platforms for component-resolved diagnostics in allergic patients. Clin Exp Allergy 2021, 51, 1603-1612.

Author Response

The manuscript deals with the development and validation of a new diagnostic molecular test for veterinary medicine, which is currently unique in the world. It has been used in three laboratories in Europe over the last two years and tested on more than 23000 dog sera. The topic is original and absolutely relevant to the field as it fills a specific gap in the field. Compared to other publications, the research described in the manuscript provides data for the development and validation of a completely new diagnostic method that cannot be compared to other diagnostic methods in the field of veterinary allergology due to the different, i.e. molecular, methods used. Older methods were all based on the ELISA test. The research is based on the methodology and the number of animals, which is sufficient to support the conclusions. The conclusions are consistent with the evidence and arguments presented and address the main question posed. References are appropriate. The tables and figures are representative.

The manuscript is well written, with a clear structure, perfectly written material and methods and presentation of results, with a structured discussion and clearly understandable conclusions after each part of the discussion. All of us who deal with allergic dogs on a daily basis, as well as researchers in the field of veterinary dermatology, can only learn from this article.

Minor notes:

Reference no. 16= Reference no. 14* and this must be corrected throughout the text, therefore:

Line 80:...« or singleplex molecular assays [13-20].« should be corrected to [13-19]. Consequently, from line 101 to line 232, all numbers for references should be reduced by one number. The citation numbers between lines 582-729 and 742-765 must also be corrected (reduced by one number).

Response: Thank you very much for catching this mistake, which is an error of the bibliographic software used. We corrected the references and added a couple of new ones published recent.y

  1. Lines 182-184: the sentence »Indeed, we had confirmed in other studies that IgE against mite tropomyosins is cross reactive with that of the Toxocara canis nematode and is thus positive in many healthy dogs, including these laboratory beagles.« needs to be referenced.

Response: We added the reference to a newly-accepted paper on this topic.

Reviewer 3 Report

Comments and Suggestions for Authors

The purpose of the present manuscript was to explain the development and validation of a new method for the detection of canine allergen-specific IgE. The article elucidates the high accuracy and internal consistency of the method, as well as the low interference by confounding factors, making this method a novel and remarkable advance in veterinary allergology.

Author Response

The purpose of the present manuscript was to explain the development and validation of a new method for the detection of canine allergen-specific IgE. The article elucidates the high accuracy and internal consistency of the method, as well as the low interference by confounding factors, making this method a novel and remarkable advance in veterinary allergology.

Response: We appreciate the kind review.

Reviewer 4 Report

Comments and Suggestions for Authors

This work demonstrates how the serological test with the Pet Allergy Xplorer (PAX), can highlight important and reliable data on environmental allergens in allergic dogs. In my opinion, the epidemiological work that this device can do with collecting the percentages of allergens involved in a large number of allergic dogs is very important; as well as defining the standard of collection, conservation and times of sending samples to have a good repeatability of the results. While it is recommended to set up a prospective study on the results (efficacy) of immunotherapy based on the results of the PAX compared to other serological tests and/or intradermal tests, in long-term allergic dogs.

Author Response

This work demonstrates how the serological test with the Pet Allergy Xplorer (PAX), can highlight important and reliable data on environmental allergens in allergic dogs. In my opinion, the epidemiological work that this device can do with collecting the percentages of allergens involved in a large number of allergic dogs is very important; as well as defining the standard of collection, conservation and times of sending samples to have a good repeatability of the results. While it is recommended to set up a prospective study on the results (efficacy) of immunotherapy based on the results of the PAX compared to other serological tests and/or intradermal tests, in long-term allergic dogs.

Response: We appreciate the suggestions for additional studies.

Line 59: The data are reported in human studies. The clinical relevance of the individual components has not been clinically tested in the allergic dog. Therefore the statement: "each allergen “spot” contains 100% of a single allergen rather than a crude mix of proteins, with few of them being relevant allergens"; must be rephrased in light of current data.

Response: The point is well taken. We started the sentence with “In humans”, to make it clear that we are writing about that species.

Line 67: The presence of specific IgE for different allergens has no correlation (no studies in dogs) between quantity and clinical significance (it cannot be associated with the percentage of clinical itching that specific allergen can cause in that single tested subject).

Response: We did not write—or imply—that there was a correlation between quantity of IgE and clincial significance. Our sentence is the following: “Other known advantages of molecular allergology are the easier identification of allergen cross-reactivity and primary sensitizing allergens, the possibility of predicting clinical evolution or clinical signs with the detection of IgE sensitization to some specific components, and a more logical selection of allergens for AIT”. This sentence is legitimate as the observation of some unique molecular sensitization profiles can predict the severity of signs and the clinical evolution, for example with peanut allergens: a sensitization to profilins and nsLTPs will lead to the oral allergy syndrome, while one to Ara h 2 will lead to anaphylaxis. This is well described in the citation provided.

Line 94: It is not clear from the text whether the selected beagle dogs are allergic and, if so, with what type of allergy profile (positive allergens panel)

Response: we added the information about the beagles, and more details are found later in the text.

Line 102: This point may be a bias of the study given the postulates on the limits of the extracts with respect to the allergenic components.

Response: We do not understand this point in response to the following sentences: “We aimed to design the PAX similarly to the ALEX2, with around one-third of allergen extracts and two-thirds of molecular components. The selection of allergens for the PAX for dogs began with those offered on the ALEX2 after postulating that if a protein is allergenic in humans, it would likely also be so in animals. “

There is no bias here, as we are just describing what was done.

Line 109: explain better if exotic fruits and shrimp are considered as environmental or food allergens. Because the limits of identifying food allergens are many and different from environmental ones (times and mechanism of action for the development of skin lesions-itching and appearance of gastro-enteric symptoms). Food substances can then cause intolerance as well as allergy.

Response: We added the words “food allergens”.

Line 263: The authors explain why, from a statistical point of view, the serum of only 4 allergic dogs was sufficient.

Response: As the reviewer can see in the result section 3.3.1, we found a significant difference between serum and plasma samples, even with just four dogs. This significant difference confirms that our study design was not under-powered. A lack of power due to small number of patients could only apply if a statistically significant difference were not seen.

Line 598: This concept of allergens below the test positivity threshold that may still be of clinical relevance must be included in the information to the vets.

Response: Such a statement is included in all results sent to veterinarians. This is not directly relevant to this paper.

Line 662: In the future, data on the stability of the samples on a larger number of dogs are required to have more consistent data.

Response: As written above, even with just four dogs, we found a significant difference between sample types and temperature storage. Nevertheless, we will take this suggestion under advisement.

Round 2

Reviewer 1 Report

Comments and Suggestions for Authors

The authors have addressed all concerns.  In my opinion, this manuscript is acceptable for publication.